# Longitudinal ultrasonic dimensions and parametric solid models of the gravid uterus and cervix

Erin Marie Louwagie[2], Lindsey Carlson[1], Veronica Over[2], Lu Mao[3], Shuyang Fang[2], Andrea Westervelt[2], Joy Vink[4], Timothy Hall[5], Helen Feltovich[1‡], Kristin Myers[2‡]*

**1** Maternal Fetal Medicine, Intermountain Healthcare, Provo, UT, United States of America, **2** Department of Mechanical Engineering, Columbia University, New York, NY, United States of America, **3** Department of Biostatistics and Medical Informatics, University of Wisconsin, Madison, WI, United States of America, **4** Department of Obstetrics & Gynecology, Columbia University Irving Medical Center, New York, NY, United States of America, **5** Department of Medical Physics, University of Wisconsin, Madison, WI, United States of America

☯ These authors contributed equally to this work.
‡ HF and KM also contributed equally to this work.
* kmm2233@columbia.edu

**Data Availability Statement:** Model files are available from the Columbia University Libraries' Academic Commons (https://doi.org/10.7916/d8-

## Abstract

Tissue mechanics is central to pregnancy, during which maternal anatomic structures undergo continuous remodeling to serve a dual function to first protect the fetus in utero while it develops and then facilitate its passage out. In this study of normal pregnancy using biomechanical solid modeling, we used standard clinical ultrasound images to obtain measurements of structural dimensions of the gravid uterus and cervix throughout gestation. 2-dimensional ultrasound images were acquired from the uterus and cervix in 30 pregnant subjects in supine and standing positions at four time points during pregnancy (8-14, 14-16, 22-24, and 32-34 weeks). Offline, three observers independently measured from the images of multiple anatomic regions. Statistical analysis was performed to evaluate inter-observer variance, as well as effect of gestational age, gravity, and parity on maternal geometry. A parametric solid model developed in the Solidworks computer aided design (CAD) software was used to convert ultrasonic measurements to a 3-dimensional solid computer model, from which estimates of uterine and cervical volumes were made. This parametric model was compared against previous 3-dimensional solid models derived from magnetic resonance frequency images in pregnancy. In brief, we found several anatomic measurements easily derived from standard clinical imaging are reproducible and reliable, and provide sufficient information to allow biomechanical solid modeling. This structural dataset is the first, to our knowledge, to provide key variables to enable future computational calculations of tissue stress and stretch in pregnancy, making it possible to characterize the biomechanical milieu of normal pregnancy. This vital dataset will be the foundation to understand how the uterus and cervix malfunction in pregnancy leading to adverse perinatal outcomes.

wxem-e863; https://doi.org/10.7916/d8-gxv7-2z02; https://doi.org/10.7916/d8-g3bz-yj53).

**Funding:** Research reported in this publication was supported by the Eunice Kennedy Shriver National Institute Of Child Health & Human Development (url: https://www.nichd.nih.gov/) under Award Number R01HD091153 to KMM and under Award Number F31HD082911 and R01HD072077 to HF and TH. The content is solely the responsibility of the authors and does not necessarily represent the official views of the National Institutes of Health. The funders had no role in study design, data collection and analysis, decision to publish, or preparation of the manuscript.

**Competing interests:** The authors have declared that no competing interests exist.

## Introduction

To date, there has been a lack of clinical, translational, and basic science research in the field of reproductive biomechanics and bioengineering. To illustrate, although parturition (labor and delivery) is so common that every human has experienced it, there are currently no clinical tools to effectively predict when delivery will happen, how long pregnancy will last, and any complications which may arise. This lack of understanding of fundamental pregnancy biomechanics makes it extremely challenging to understand and address abnormal pregnancy conditions such as preterm birth (PTB, delivery before 37 weeks gestation), which affects 10% of deliveries worldwide and carries short- and long-term health consequences from death in the neonatal period to lifelong disability [1].

The mechanical integrity and function of reproductive tissues is clearly critical to pregnancy outcome [2–5]. The uterus, fetal membranes, and cervix each have dynamic, biological, and mechanical roles (Fig 1); these tissues must remodel and stretch to accommodate the growing fetus while it develops *in utero*, and then do the opposite, i.e. contract, deform, or rupture, to facilitate safe delivery of the fetus. Failure and mistiming of these essentially mechanical events contribute to major obstetrical complications such as PTB [6, 7].

The vital knowledge gap in fundamental pregnancy physiology exists in part because it is challenging to obtain direct quantitative data on how the uterus, fetal membranes, and cervix change throughout pregnancy as pregnancy is a protected environment. This is why we propose a biomechanical parametric modeling approach. Our ultimate goal is to facilitate precision medicine for parturition via development of personalized computational models to

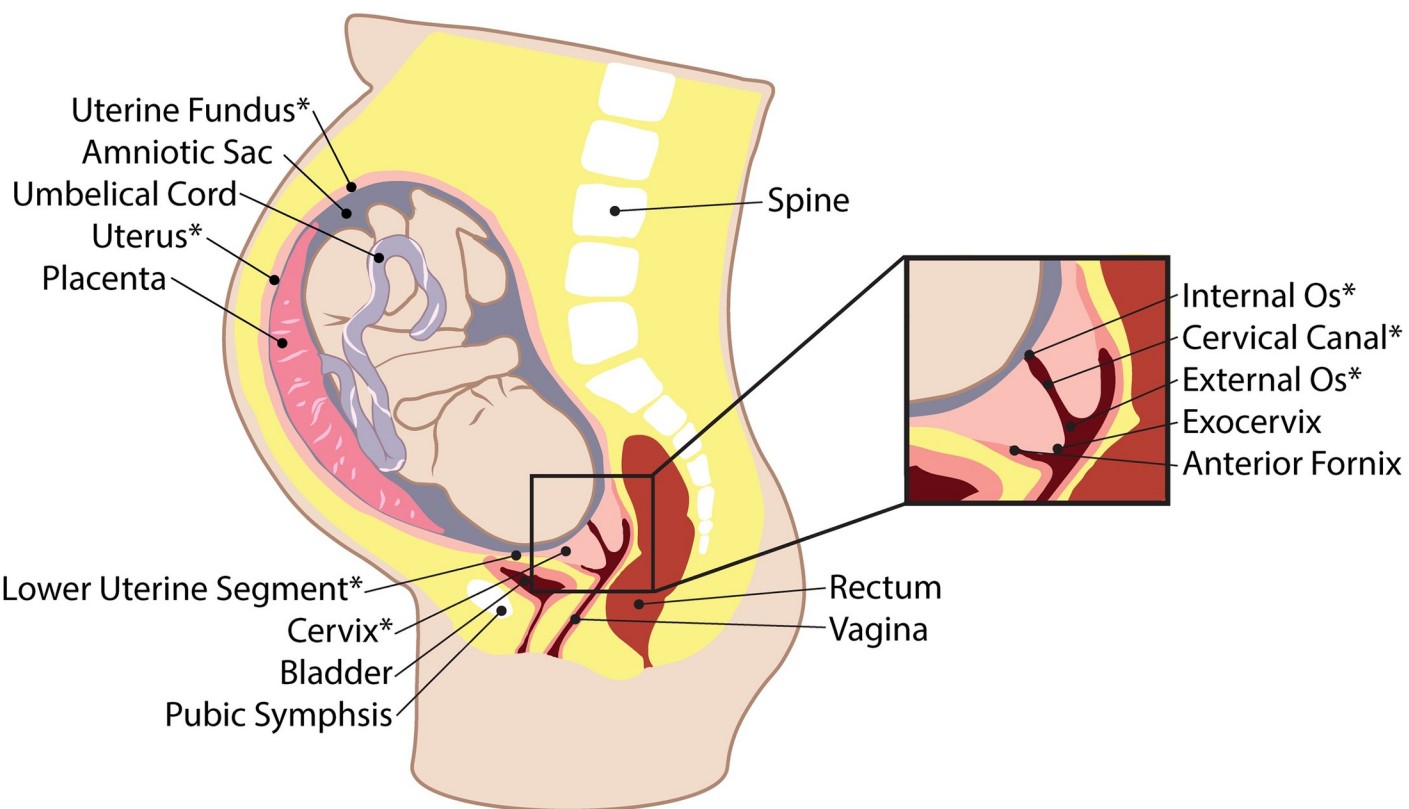

**Fig 1. Pregnant anatomy.** Representative illustration of a sagittal view of pregnant anatomy with relevant reproductive and surrounding structures labeled. Asterisks (*) indicate structures evaluated in the protocol.

characterize a patient-specific biomechanical environment in pregnancy. As a step toward that, the goal of the present work is to provide time-course maternal anatomy data and corresponding 3-dimensional computer aided design (CAD) models on a cohort of low-risk patients with normal singleton pregnancies.

We acquired imaging data with a standard clinical ultrasound imaging system for practical reasons because compared to other imaging modalities, ultrasound is relatively inexpensive, convenient, and low risk. Fortunately, we found it is feasible to use 2D images to obtain accurate measurements of maternal anatomy to create a CAD model for comprehensive visualization of maternal anatomy. Here, we report: 1) values of critical anatomic structures in normal gestation based on images from quick 2D ultrasound data acquisitions, 2) reproducibility and reliability of each individual measurement and its value to the overall model, 3) effect of gestational age, gravity, and parity on maternal geometry 4) corresponding simplistic and robust 3D parametric CAD models of the uterus and cervix (Solidworks, Dassault Systémes, Vélizy-Villacoublay, France), 5) estimates of time-course uterine and cervical volumes throughout pregnancy, and 6) a comparison study of the parameterized 3D solid model to MRI-derived solid models.

The data and models generated in this study establish a quantitative foundation for computational analysis of pregnancy. Additionally, the 3D solid modeling method provides the critical foundation to understand how these reproductive tissues may malfunction in pregnancy and allow for novel avenues to design biomedical devices which can be used to prevent adverse outcomes such as PTB.

## Materials and methods

Two-dimensional (2D) ultrasound acquisitions from the uterus and cervix were taken from 30 women at four time points (8w0d-13w6d, 14w0d-16w6d, 22w0d-24w6d, and 32w0d-34w6d gestation) during pregnancy and stored for offline measurements of dimensions of the uterus and cervix. Reproducibility and reliability of each parameter and its potential contribution to the model was assessed and correlations between parity, maternal age, and gestational age were evaluated. Parameterized patient-specific models were built using Solidworks for all patients and time points using the validated dimensions. Estimates of uterine and cervical tissue volume were determined from these models. Shape parameterization effects were explored by applying the 2D ultrasound measurement protocol to, and comparing the resulting parametric model with, segmented MRI solid models.

### Study design

This was a longitudinal study of ultrasound dimension measurements in women at low-risk for preterm birth at 8w0d-13w6d, 14w0d-16w6d, 22w0d-24w6d, and 32w0d-34w6d gestation.

### Patients

Flyers describing this study were given to low-risk obstetric clinics whose providers deliver and refer to Utah Valley Hospital in Provo Utah. Interested patients called L.C.C., who reviewed eligibility criteria with them. Thirty patients ages 18-41 were recruited when they were in the 1st trimester (<14 weeks gestation) from Valley Womens Health in Provo, Utah from July to December of 2017. Exclusion criteria included history of preterm birth, prior cesarean delivery for failure to progress in labour (failure of the cervix to soften/dilate), previous cervical surgery (including cerclage/LEEP/cone), collagen vascular disease, or known uterine malformation. Inclusion criteria included singleton pregnancy in the first trimester and maternal age 18-45 years old. This study was approved by the institutional review boards at

Intermountain Healthcare and the University of Wisconsin, and each subject provided written informed consent.

The age, race, ethnicity, pregnancy history, estimated gestational age (EGA) at each visit, EGA at delivery, and delivery outcomes were recorded for each patient. Details about this cohort, along with study size rationale, are published in Carlson et al. [8]. The estimated date of delivery (EDD) was confirmed by ultrasound crown-rump length at the first study visit. One patient delivered preterm (at 34 weeks 5 days) and was therefore excluded from analysis. Of the remaining 29 patients included in the final analysis, 9 were nulliparous (first pregnancy) and 20 multiparous (at least 1 previous delivery). Of these, none of the pregnancies were considered high-risk and no pregnancy issues were found. Patient demographics are reported in Table 1.

## Ultrasound data acquisition

All ultrasound examinations were done by the same sonographer (J.D.) and acquisitions overseen by the same engineer (L.C.C.). Scanning was performed using a Siemens ACUSON S3000 ultrasound system (Siemens Healthcare, Ultrasound Business Unit, Mountain View, CA, US). The designated research sonographer (J.D.) was certified for cervical length measurement through the Perinatal Quality Foundation's Cervical Length Education and Review (CLEAR) program (url:perinatalqualityfoundation.org). Transabdominal measurements followed the protocol established by Saul et al. 2008 [9].

Each participant underwent ultrasound an exam at four different time points: 1st trimester (8w0d-13w6d gestation), early 2nd trimester (14w0d-16w6d gestation), mid 2nd trimester (22w0d-24w6d gestation), and 3rd trimester (32w0d-34w6d gestation). The rationale for performing two evaluations during the 2nd trimester is that 16-24 weeks currently appears to be the most critical period for preterm birth-associated cervical change in pregnancy [6]. During each visit, six B-mode ultrasound images of the uterus and cervix were acquired (Figs 2–4), three with the patient supine and three standing. These included transabdominal (TA) sagittal views of the uterus and cervix, a TA axial view of the uterus, and a transvaginal (TV) sagittal view of the cervix and lower uterine segment. Patients emptied their bladders prior to scanning.

TA sagittal and axial scans were acquired using the SieScape extended field-of-view imaging feature on the ultrasound system, which automatically registers adjacent images together as the transducer is swept across the abdomen. The resulting panoramic image is not dependent on the speed of image acquisition. Examples of this extended field-of-view imaging in the sagittal and axial views for a participant at 32 weeks are shown in Figs 2 and 3, respectively. For TV acquisitions, the transvaginal transducer was placed into the anterior fornix of the vagina, the image optimized, and the landmarks identified (internal and external os, canal). All measurements of the dimension parameters were taken from deidentified ultrasound images ImageJ Software [10] (NIH, Besthesda, MD). A representative selection of images from the first few patients was used by L.C.C. to instruct the two research sonographers on making

**Table 1. Patient demographics including the number of subjects, N, in each group and their average age (age range in parentheses).** There were no patients with a BMI larger than 30.

|  | N | Age | Previous Cesarean Section |
|---|---|---|---|
| Nulliparous | 9 | 27 (22–30) | - |
| Multiparous | 21 | 28 (21–37) | 3 |
| Total | 30 | 28 (21–37) | 3 |

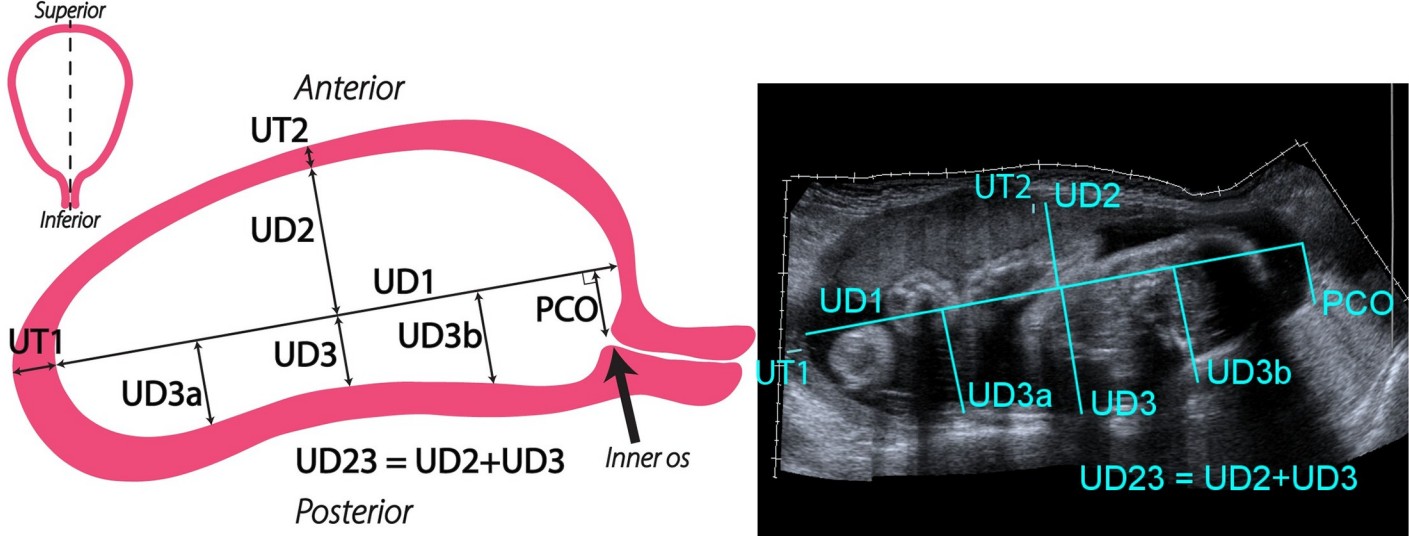

**Fig 2. Representative and actual transabdominal (TA) sagittal scan of a pregnant patient at 32 weeks.** Left: coronal uterine outline with the ultrasound sweep location shown with a dashed line, and a representative illustration of measurements taken from ultrasounds of the uterus and cervix in the sagittal view. Right: actual transabdominal (TA) sagittal view of the uterus taken as a extended field-of-view ultrasound sweep from uterine fundus to the cervix. Measurements taken from the TA sagittal view are: inferior-superior (UD1) and anterior-posterior intrauterine dimensions (UD2, UD3, UD3a, UD3b), uterine wall thickness measurements (UT1, UT2), and the distance the internal os is offset from the inferior-superior uterine axis (PCO).

measurements. The 3 research team members then independently recorded measurements on the entire dataset.

The 16 parameters describing dimensions of the uterus and cervix were based upon previous work [11]. From TA sagittal images, the inferior-superior intrauterine diameter (UD1) was measured as the longest dimension from the fundus to the lower uterine segment (Fig 2). From the midpoint of UD1, the perpendicular distance to the anterior (UD2) and posterior (UD3) intrauterine walls were measured. To quantify the position of the cervix in relation to

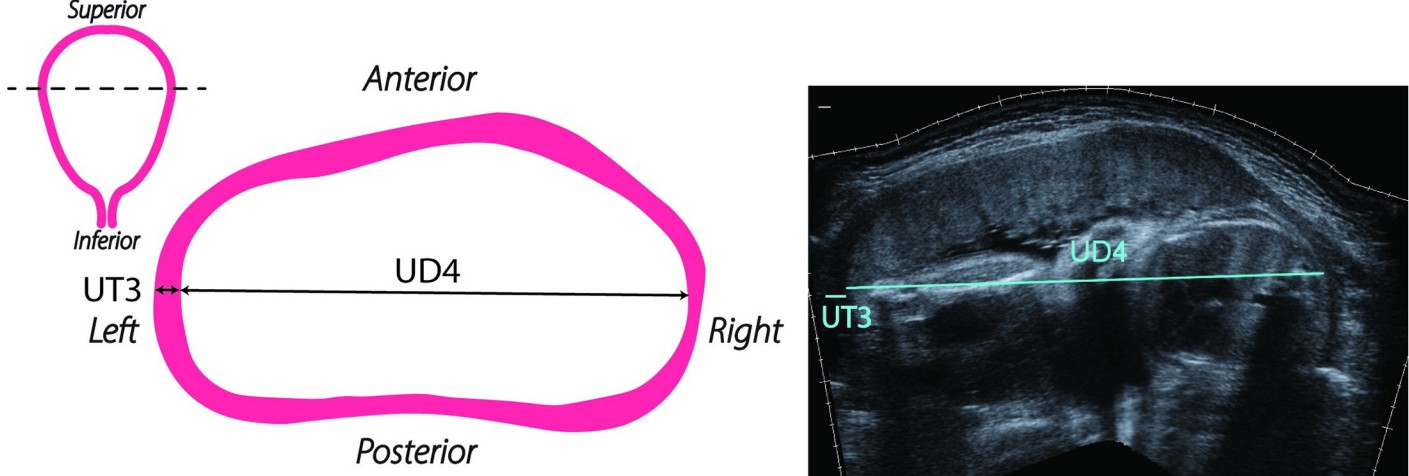

**Fig 3. Representative and actual transabdominal (TA) axial scan of a pregnant patient at 32 weeks.** Left: coronal uterine outline with the ultrasound sweep location shown with a dashed line, and a representative illustration of measurements taken from ultrasounds of the uterus in the axial view. Right: actual transabdominal (TA) axial view of the uterus taken as a extended field-of-view ultrasound sweep from left to right at the widest section of the uterus. Measurements taken from the TA axial view are either left or right uterine wall thickness (UT3) and left-right uterine diameter (UD4).

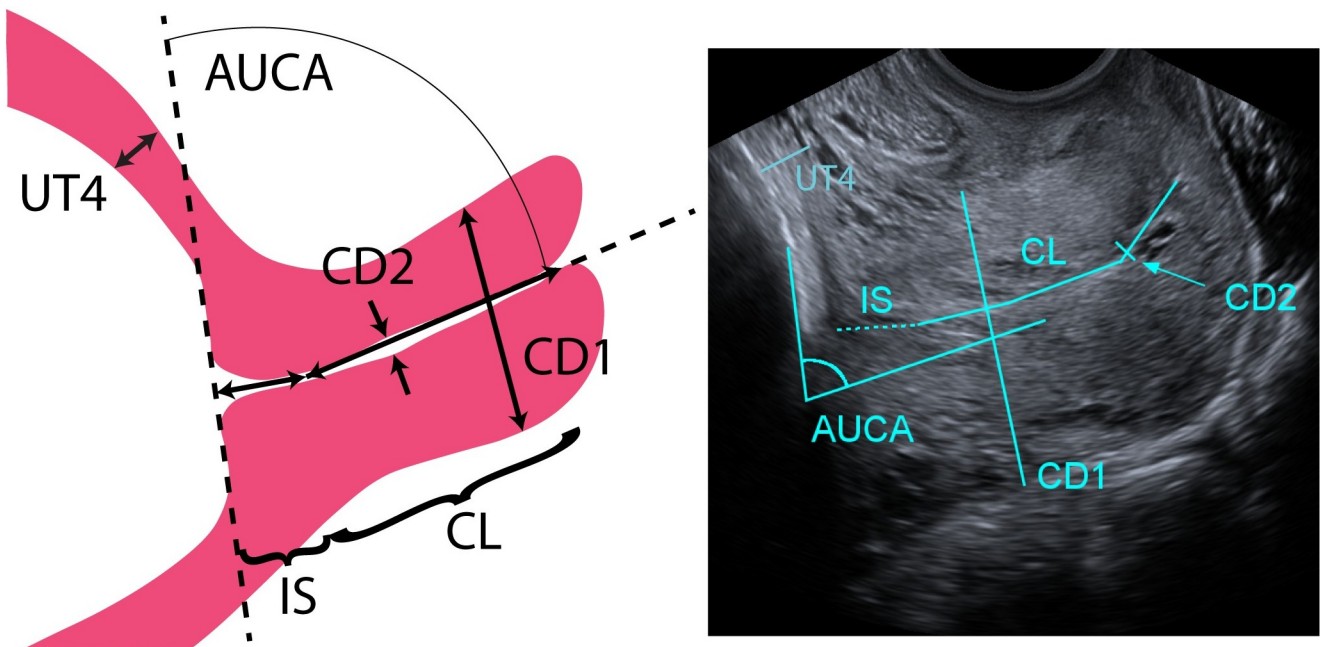

**Fig 4. Representative and actual transvaginal (TV) sagittal scan of a pregnant patient at 32 weeks.** Left: representative illustration of measurements taken from ultrasounds of the cervix and lower uterine segment in the sagittal view at the maternal midline. Right: actual transvaginal (TV) sagittal view of the cervix and lower uterine segment taken by placing the transvaginal ultrasound probe on the anterior fornix and turning the probe to view the sagittal plane. Measurements taken in the TV sagittal view are: cervical length (CL), outer diameter (CD1), inner cervical canal diameter (CD2), and cervical angle with anterior wall of the uterus (AUCA).

the uterus, the perpendicular cervical offset (PCO) distance of the cervical internal os to UD1 was measured. Additional posterior dimensions (UD3a & UD3b) perpendicular to UD1 were taken at 25% and 75% of UD1 from the superior intrauterine wall, respectively, to describe the curvature of the posterior uterine wall due to the boundary of the spine. Uterine wall thicknesses at the fundus (UT1) and anterior uterine wall (UT2) were also measured in the TA sagittal scan. However, if UT1 and UT2 were not clear in the TA sagittal scan, additional ultrasound images were taken at the specific location of the unclear measurement.

From the TA axial images (Fig 3), left-right uterine diameter (UD4) and uterine wall thickness at either the left or right wall (UT3) were measured. The TA axial scan was done at the widest section of the uterus, where UD4 represents the largest left-right axial intrauterine diameter. Again, if UT3 was not clear in the TA axial scan, then an additional ultrasound was taken at the left or right wall to obtain the wall thickness measurement. Left and right wall thicknesses were assumed to be the same.

From the TV images, uterine wall thickness at the lower uterine segment (UT4), cervical length (CL), cervical outer diameter (CD1), inner cervical canal diameter (CD2), and the anterior utero-cervical angle (AUCA) were measured (Fig 4). Care was taken to exclude from the CL the isthmus (IS), where the cervical mucosa ends [12]. During the TV exam, quantitative ultrasound data were also acquired from the cervix to measure tissue softness, as published in Carlson et al [8], for future integration into our models of information about tissue microstructure.

## Statistical analysis

To assess reproducibility (interobserver variability) and reliability of the measurements, the inter-class correlation coefficient (ICC) was estimated (p-value is for testing $H_0$: ICC = 0). The

quality of each parameter was categorized according to the Cicchetti (1994) Guideline as follows: poor (less than 0.40), fair (between 0.40-0.59), good (between 0.60-0.74), and excellent (between 0.75-1.00) [13].

A linear mixed effects model (LMM) was used to estimate the relationship between each parameter and gestational age (continuous variable *GA* in weeks), position (categorical predictor *Pos*; supine = 1, standing = 0) and parity (categorical predictor *Par*; nulliparous = 1, multiparous = 0) and fitted using maximum likelihood. Random effects due to multiple observers and uncorrelated random effects due to intersubject variability for slope and intercept were included. An LMM model for a measurement can be represented as follows:

$$\text{Meas}_i = a' + b' * \text{GA} + c' * \text{Pos} + d' * \text{Par} + e' * \text{GA} * \text{Pos} + f' * \text{GA} * \text{Par} \tag{1}$$

where primed (') variables include random effects. For each model, 95% confidence intervals were estimated for the fixed effects via parametric bootstrapping (10,000 iterations), and approximate p-values were subsequently found via inversion of estimated confidence intervals. Statistical analysis was performed in R version 3.3.2 (R Core Team, 2014, R Foundation for Statistical Computing, Vienna, Austria; available at: http://www.R-project.org/).

## Parametric CAD model

Solidworks 2018-19 (Dassault Systémes, Vélizy-Villacoublay, France) was used to construct solid models of the uterine and cervical geometries. A *design table* approach was used to allow for automatic generation of patient specific geometries based on the list of anatomical dimensions described in Figs 2–4. To establish a parametric build workflow, a *Default Configuration* was established where geometric relations for all subsequent models are established (Fig 5). Detailed information of the Solidworks workflow is presented in S1 Appendix and a video created and recorded for the 2020 Summer Biomechanics, Bioengineering, and Biotransport Conference (SB3C), held online June 17-20, 2020 (available: https://doi.org/10.7916/d8-wxem-e863) [14].

## Volume measurement

The parametric solid model created in Solidworks was used to estimate uterine and cervical volume over the course of gestation for all patients in both the supine and standing positions. Because the parametric solid model creates the uterus and cervix as a single part, the cervical volume had to be separated from the uterine volume. CD1 was selected as the uterocervical boundary. The sketch of the outer cervical diameter (CD1) was made into a surface using the *Extended Surface* tool at a distance equal to the cervical length, and the *Split* feature used to separate the cervix from the uterus. The *Mass Properties* tool in Solidworks was used to calculate the volume of the uterus and cervix individually.

## Validation of parametric model

MRI data from Joyce et al. were obtained for 8 term pregnant women prior to caesarean delivery [4]. Patient age ranged from 32 to 47 with a mean gestational age of 38.41 ± 0.36 weeks. MRIs were taken within 0-7 days of the scheduled delivery. The MRI image stacks were then segmented using the commercial software package Materialise Mimics (Research 20.0, Materialise MV, Leuven, Belgium). The detailed protocol of model builds for validation and method for model comparison are in S2 Appendix.

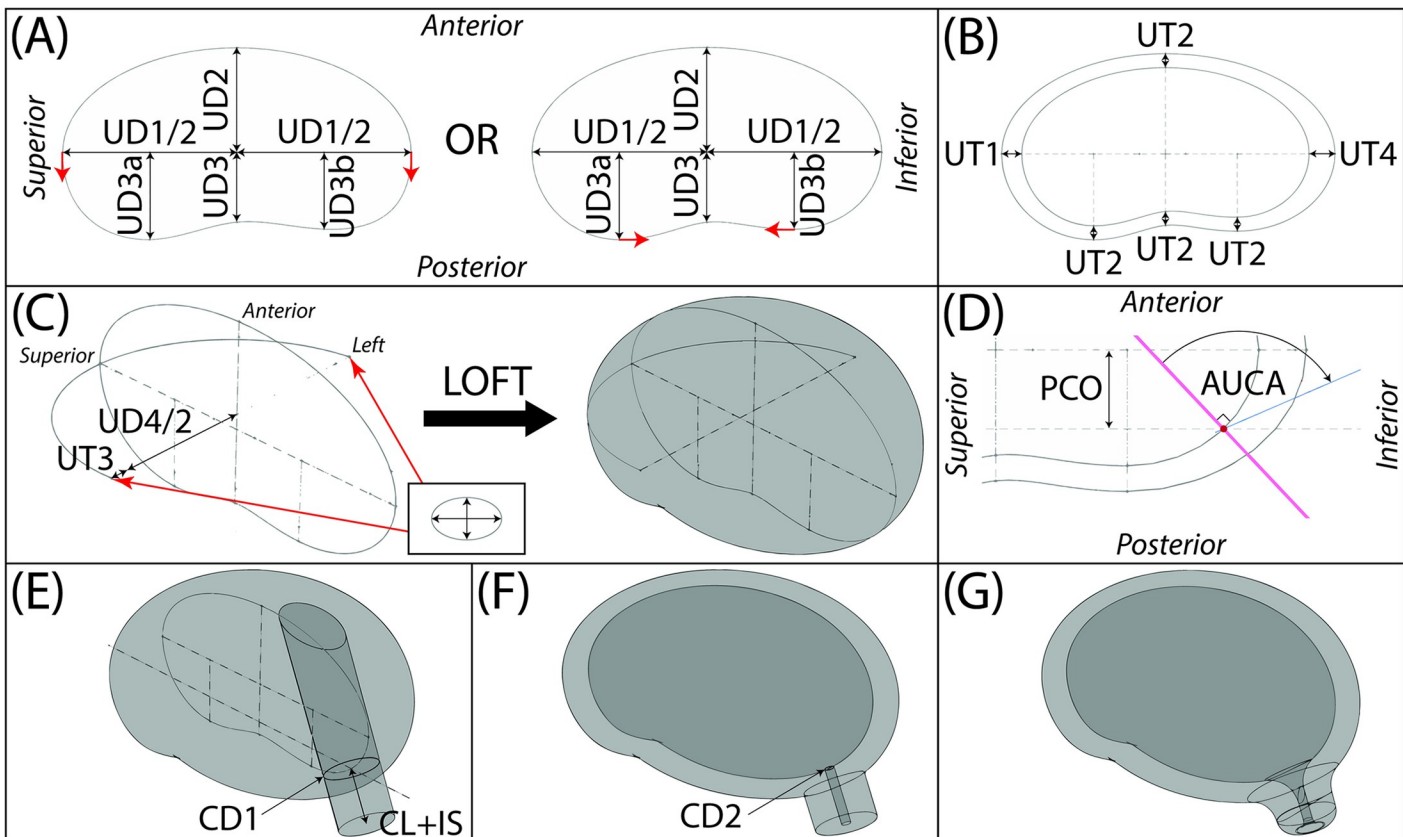

**Fig 5. CAD model construction.** (A) Inner uterine wall sagittal shape built from ultrasound measurements. Posterior side is built with either a spline (left) or ellipses and a spline (right). Red arrows indicate where tangency constraints have been applied. (B) Outer uterine wall sagittal shape built by adding uterine wall thickness ultrasound measurements to inner uterine wall sagittal shape. (C) Outer uterine loft completed using scaled ellipses as left and right profiles and a half ellipse as the guide curve. (D) Internal os is placed at a distance of the perpendicular cervical offset (PCO) from the inferior-superior intrauterine diameter (UD1) on the posterior wall and a plane at an angle of the anterior uterocervical angle (AUCA) to the perpendicular of the posterior wall. (E) Cervix is added by extruding a circle with diameter of the outer cervical diameter (CD1) away from the outer uterine loft a length of the combined cervical and isthmus lengths (CL+IS) and up to the outer surface of the uterine loft. (F) Intrauterine cavity is made using a lofted cut and the cervical canal is added by an extruded cut to the end surface of the cervix and the surface of the intrauterine cavity. (G) Fillets are added at the internal os, external os, exocervix, and uterocervical junction.

## Results and discussion

Overall uterine diameters collected via transabdominal ultrasound and dimension measurements collected via transvaginal ultrasound have excellent and good agreement between observers. As expected, all uterine diameters increase with gestation, while lower uterine segment thickness (UT4) and isthmus length (IS) decrease. The anterior uterocervical angle shifts posteriorly with gestation. Parity influences the lower uterine segment thickness (UT4) and cervical length (CL) measurement trends. Multiparous patients have a greater rate of lower uterine segment thinning, compared to nulliparous patients, and cervical lengths that remain constant with gestation. Nulliparous patients' cervical length decrease throughout gestation. Maternal position, supine vs. standing, was not significant for lower uterine segment thickness (UT4), cervical length (CL), and isthmus length (IS). However, maternal position affected uterine diameters (UD1-UD4, UD3a, and UD3b) and anterior uterocerivcal angle (AUCA). The parametric solid modeling method is able to automatically generate models based on patient-specific dimension measurements in 91% of cases. All ultrasonic dimension data and corresponding solid models are available at Columbia

University's Academic Commons (dimensions: https://doi.org/10.7916/d8-g3bz-yj53, models: https://doi.org/10.7916/d8-tchz-hs47).

## Ultrasound parameters

**Reproducibility and reliability.** The majority of intrauterine diameter measurements (UD1-4, UD3a) and lower uterine wall thickness measurement (UT4) showed excellent agreement between observers (Table 2, ICC>0.75). The inferior perpendicular distance from the inferior-superior axis to the posterior intrauterine wall (UD3b) showed good agreement between observers (0.60<ICC<0.74). Good agreement was also demonstrated for cervical length (CL), isthmus length (IS), and anterior uterocervical angle (AUCA). Fair agreement between observers (0.40<ICC<0.59) was noted for the outer cervical diameter (CD1) and poor agreement (ICC<0.40) was seen for uterine wall thicknesses at the fundus, anterior uterine wall, and left or right uterine wall (UT1-UT3), along with the posterior cervical offset (PCO) and inner cervical canal diameter (CD2). Parameters with fair or poor agreement were removed from further analysis. The ICC values for each measurement are summarized in the last column in Table 2.

The intrauterine diameter measurements accounting for the posterior uterine wall at 25% and 75% along the inferior-superior axis (UD3a and UD3b) show the lowest ICC values of all intrauterine diameters, likely because of difficulty in viewing posterior features in the TA sagittal view. The inconsistency in image quality also precludes measurement of posterior uterine wall thicknesses, and in models that posterior wall thickness is assumed to be equal to the anterior wall thickness (UT2). As expected, the sum (UD23) of the perpendicular distance from the midpoint of the inferior-superior intrauterine diameter (UD1) to the anterior (UD2) and posterior (UD3) intrauterine wall shows a higher ICC value than the measurements individually, likely because it spans the entire anterior-posterior intrauterine diameter, making it independent of the placement of UD1.

**Table 2. Summary of linear mixed model and ICC results for each parameter sorted by highest to lowest ICC value.** LMM slope estimates (mm/week) in linear mixed effects models adjusting for parity and position (supine versus standing). The line indicates the cutoff between good and fair measurements, as prescribed in the methods section. *Indicates measurements for which parity was significant and included in the model.

| Measurement | b (LMM slope) [mm/wk] | 95% CI | P-value | ICC |
|---|---|---|---|---|
| UD1 | 8.831 | (8.75–8.91) | <0.001 | 0.990 |
| UD4 | 7.912 | (7.81–8.01) | <0.001 | 0.984 |
| UD23 | 2.795 | (2.74–2.85) | <0.001 | 0.936 |
| UD3 | 1.753 | (1.71–1.80) | <0.001 | 0.853 |
| UD2 | 1.060 | (1.02–1.00) | <0.001 | 0.827 |
| UT4* | -0.324 | (-0.35– -0.30) | <0.001 | 0.823 |
| UD3a | 2.013 | (1.96–2.06) | <0.001 | 0.820 |
| UD3b | 1.353 | (1.31–1.40) | <0.001 | 0.739 |
| AUCA (Deg./wk) | 0.369 | (0.24–0.50) | <0.001 | 0.691 |
| IS | -0.436 | (-0.47– -0.40) | <0.001 | 0.677 |
| CL* | -0.065 | (-0.09– -0.04) | <0.001 | 0.625 |
| CD1 | 0.270 | (0.25–0.29) | <0.001 | 0.562 |
| PCO | 0.632 | (0.58–0.68) | <0.001 | 0.365 |
| UT1 | 0.029 | (0.02–0.04) | <0.001 | 0.263 |
| UT2 | -0.013 | (-0.02–0.00) | 0.023 | 0.230 |
| UT3 | 0.055 | (0.04–0.07) | <0.001 | 0.202 |
| CD2 | 0.011 | (0.01–0.02) | <0.001 | 0.076 |

The significantly higher ICC value for the lower uterine segment thickness (UT4) as compared to the other uterine wall thicknesses parameters (UT1-UT3) is undoubtedly due to the use of TV ultrasound to obtain this image. As compared to TA transducers, TV transducers provide better image resolution because they operate at a higher frequency [15], they acquire data directly from the structure instead of having several tissue layers to penetrate, and the image covers a much smaller area so the features appear larger, all of which contribute to a more precise measurement. The ICC value for uterine thickness measurements captured using TA transducers could be improved by taking taking individual, zoomed in ultrasound images of the uterine wall at the fundus, anterior wall, and left or right wall.

Besides uterine thickness measurements, several other parameters showed fair or poor agreement between observers: PCO, CD1, and CD2. The poor agreement between observers for posterior cervical offset (PCO) measurements may also be attributed to variable inferior-superior intrauterine diameter (UD1) placement, as it is used as the end point for this dimension. Additionally, identification of the internal cervical os is often difficult in TA sagittal scans (this is why the clinical gold standard for measurement of the cervix is TV), most likely the primary cause for poor agreement on the PCO parameter. The fair agreement for the outer cervical diameter (CD1) is likely a result of inadequate measurement definition; the location along the cervix to measure the diameter was not specified. The poor agreement between observers for the inner cervical canal diameter (CD2) can be attributed to inadequate visualization of the cervical canal in some images, and the small magnitude of the measurement (single pixel differences can have large effects on the measurement value). This finding is consistent with previous reports describing characterization of the mucous plug, which is assumed to fill the inner cervical canal diameter [16].

Previously studies report isthmus length (IS) and anterior uterocervical angle (AUCA) to be considered repeatable measurements, while cervical length (CL) repeatability varies [17–20]. These reports are in accordance with the good agreement found for IS and AUCA, and the good agreement between sonographers for CL is most likely due to their uniform training and certification.

**Effect of gestation.**   All intrauterine diameter measurements (UD1-UD4, UD3a, UD3b) significantly increase with gestational age and cervical/lower uterine segment measurements (UT4, IS) significantly decrease with gestational age in both supine and standing positions (p-values < 0.001). The cervical length (CL) slightly decreases throughout gestation (p-value < 0.001), as has been previously described in normal pregnancy [21], and the anterior uterocervical angle (AUCA) shifts posteriorly throughout gestation (p-value < 0.001). Plots of each measurement vs. gestational age are shown for all patients in the supine position averaged across observers in S3 Appendix.

The fixed effect coefficients [*a*, *b*, *c*, *d*, *e*, *f*] for each variable [*GA*, *Pos*, *Par*, *GA* * *Pos*, *GA* * *Par*] in Eq 1 are summarized in Table 3. In the LMMs (Eq 1), the variables used with parity and position (*Pos*, *Par*) are binary, thus the coefficients that include parity and position are only applied if the subject was in the supine position and/or nulliparous. To illustrate, if a patient is in the supine position, *Pos* = 1 and variables *c* and *e* are included in the effective LMM (Eq 1). However, if a patient is in the standing position, *Pos* = 0 and variables *c* and *e* will not be included in the effective LMM (Eq 1), as they have been multiplied by 0. The same is true for position.

The dramatic increase of the uterine diameter over the course of gestation is expected, as the uterine cavity must expand to accommodate the growing fetus. The decrease in the isthmus length (IS) and lower uterine segment thickness (UT4) is also an expected finding, due to normal remodeling throughout gestation [18, 22, 23].

**Table 3. Summary of linear mixed model fixed effect coefficients from Eq 1 sorted by highest to lowest ICC value (non-significant terms are dropped with coefficients indicated as –).** The line indicates the cutoff between good and fair, as prescribed in the methods section. Measurements are based on ICC values in Table 2. Int. = intercept.

| Measurement | a<br>Int.<br>mm | b<br>GA<br>mm/wk | c<br>Pos<br>mm | d<br>Par<br>mm | e<br>GA*Pos<br>mm/wk | f<br>GA*Par<br>mm/wk |
|---|---|---|---|---|---|---|
| UD1 | -24.865 | 8.976 | 3.575 | – | -0.289 | – |
| UD4 | -30.559 | 8.205 | 17.216 | – | -0.585 | |
| UD23 | 34.666 | 2.964 | -6.830 | – | -0.110 | – |
| UD3 | 15.344 | 1.809 | -4.884 | – | -0.338 | – |
| UD2 | 18.739 | 1.193 | -1.325 | – | -0.266 | – |
| UT4 | 17.356 | -0.342 | – | -3.427 | – | 0.065 |
| UD3a | 6.280 | 2.079 | -4.703 | – | -0.133 | – |
| UD3b | 18.071 | 1.407 | -0.967 | – | -0.108 | – |
| AUCA (deg) | 73.432 | 0.014 | -5.860 | – | 0.709 | – |
| IS | 22.847 | -0.436 | – | – | – | – |
| CL | 31.759 | 0.026 | 0.577 | 3.772 | – | -0.325 |
| CD1 | 28.281 | 0.210 | -0.589 | – | 0.120 | – |
| PCO | 6.322 | 0.711 | 4.419 | – | -0.158 | – |
| UT1 | 6.665 | 0.029 | -0.724 | – | – | – |
| UT2 | 7.527 | -0.013 | -0.791 | – | – | – |
| UT3 | 7.673 | 0.055 | -0.703 | – | – | – |
| CD2 | 3.169 | 0.011 | – | – | – | – |

**Effect of parity.** Parity influences the lower uterine segment thickness (UT4) and cervical length (CL) measurements (Fig 6). UT4 decreases by 0.277 mm/wk for nulliparous patients and 0.342 mm/wk for multiparous patients. CL decreases by 0.299 mm/wk for nulliparous patients, but for multiparous patients CL stays nearly constant (small increase of 0.026 mm/wk).

The dependence on parity suggests possible permanent mechanical and structural changes that occur during the remodeling events of pregnancy. For a multiparous patient, the increased rate of thinning of the lower uterine segment suggests two possible mechanisms: 1) the mechanical load exerted by the contents of the amniotic sac is shifting faster towards the lower part of the uterus and/or 2) uterine tissue becomes softer in subsequent pregnancies. As for a multiparous cervix, there is not enough evidence in the literature to statistically determine if the cervix becomes mechanically softer with each pregnancy. One study found that women with a history of previous vaginal deliveries have softer cervices than nulliparous women [24].

**Effect of position.** Maternal position (supine vs. standing) also influences maternal geometric measurements. Maternal position vs. gestation age interaction (column 6 in Table 3) is significant for all intrauterine diameters (UD1-UD4, UD3a, and UD3b) and the anterior uterocervical angle (AUCA) where the LMM slopes are higher in the standing position, except for AUCA, where the angle decreases with standing. Maternal position is not significant for lower uterine segment thickness (UT4), cervical length (CL), and isthmus length (IS).

The effect of gravity has long been a curiosity in the study of pregnancy biomechanics. Bedrest has been demonstrated to be ineffective at reducing the rate of preterm birth [25]. The maternal anatomy measurements here confirm the cervix does not further deform when a woman stands from a supine position, nor does the lower uterine segment thin. Whether the cervix deforms after longer periods of standing (i.e. viscoelastic creep) remains to be determined. However, uterine shape does change with position. Specifically, the uterus becomes

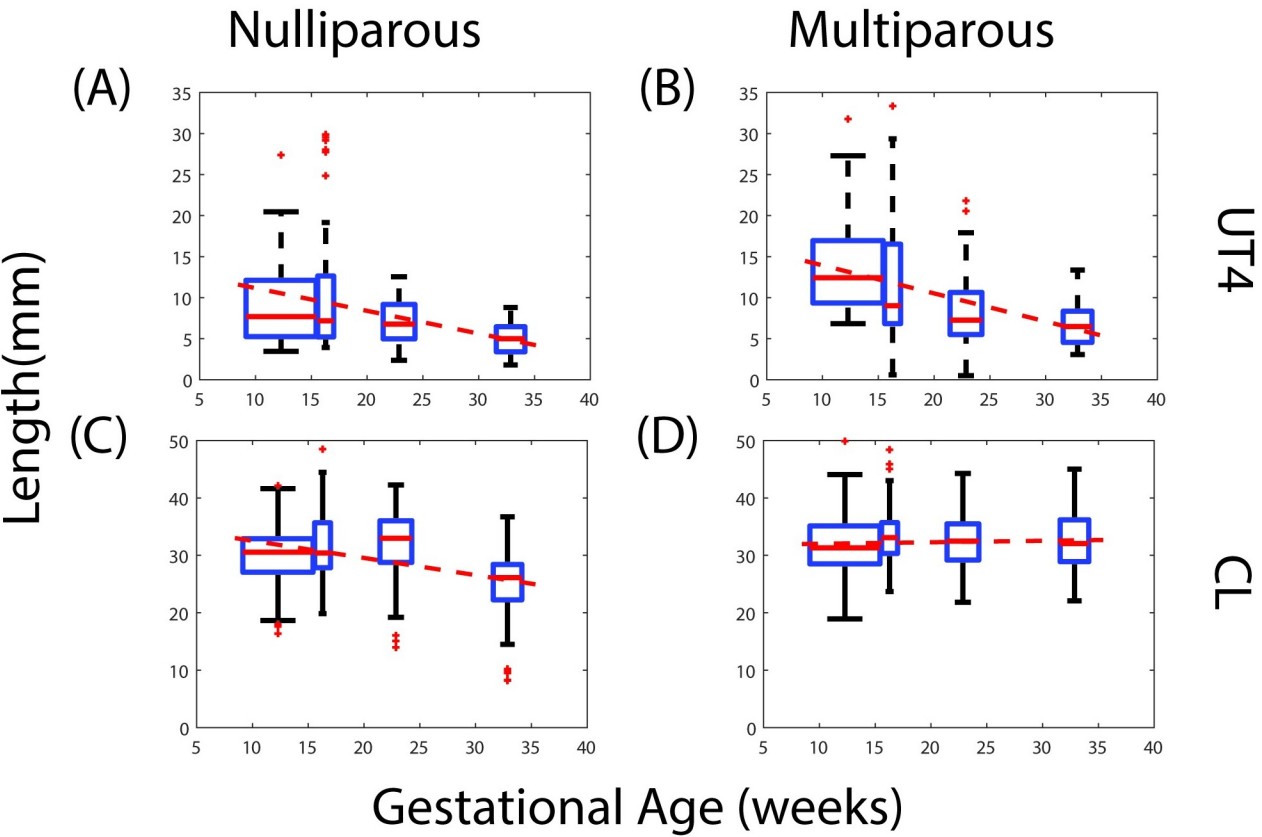

**Fig 6. Effect of parity on ultrasonic maternal anatomy measurements across gestational age.** Box and whisker plots for lower uterine segment thickness (UT4) and cervical length (CL) for nulliparous and multiparous patients.

flatter in the anterior-posterior direction and wider in the left-right direction when in the supine position when compared to the standing position. This is quantitatively observed by comparing the ratio of the anterior-posterior intrauterine diameter (UD23) to the left-right intrauterine diameter (UD4) in the standing and supine position, where in 86% of cases the ratio is larger when standing than in supine. Therefore, gravity does have an effect on uterine axial shape.

## Parametric CAD model

The solid CAD models provide a visualization of uterine and cervical shape and size change throughout gestation (Fig 7) and provide a structural foundation to calculate the mechanical loading environment of pregnancy. All solid models (STL files) generated from the workflow described in S1 Appendix are freely available through the Columbia University Library's permanent Academic Commons collection (url: https://doi.org/10.7916/d8-tchz-hs47). With 29 patients scanned at 4 time points in two positions measured by 3 sonographers, 696 sets of parametric measurements were taken and used to build models. For visits 1-3 (8w0d-13w6d, 14w0d-16w6d, and 22w0d-24w6d gestation), the *spline method* better represents the posterior uterine wall, and for visit 4 (32w0d-34w6d) the *quarter ellipse method* is a better method (see S1 Appendix for method description). Of the 696 patient-specific parametric model builds attempted, 632 usable models are generated (91% automatic build rate). Of these models, 70 require slight edits, such as altering the fillet type or radii.

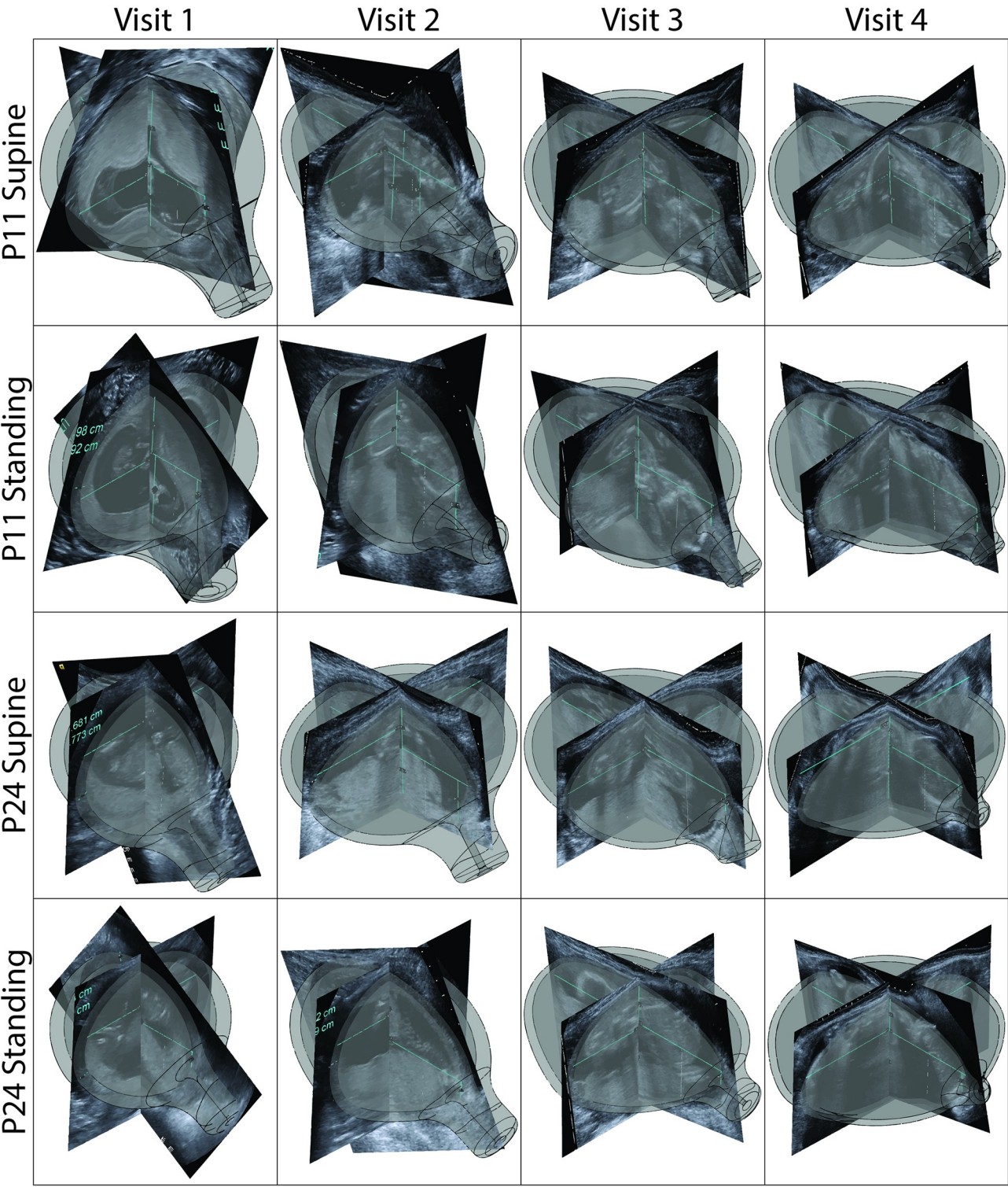

**Fig 7. Representative overlays of Solidworks model and ultrasound for all visits.** Solidworks models aligned with corresponding ultrasound scan along the inferior-superior intrauterine diameter (UD1) for (A) patient 11 in the supine position, (B) patient 11 in the standing position, (C) patient 27 in the supine position, and (D) patient 27 in the standing position for visits 1-4 (8w0d-13w6d, 14w0d-16w6d, 22w0d-24w6d, and 32w0d-34w6d gestation). Patients have been selected randomly from those where all models generated.

Cases failing to generate usable models have issues in four categories: an extreme anterior uterocervical angle (51 cases), a posterior cervical offset (PCO) larger than posterior intrauterine diameters (7 cases), or loft function failure (6 cases). Cases with an extreme anterior uterocervical angle (AUCA) fail because the cervical cylinder does not terminate correctly on the uterine body. Of these cases, 45% were visit 1, 23% were visit 2, 14% were visit 3, and 18% were visit 4. As observed in ultrasound, the CL is not typically a straight line and is frequently measured using several segments, especially early in pregnancy. Therefore, in order to model these cases, the curvature of the cervix may need to be captured. For cases where the loft function failed, Solidworks is not able to complete the loft for the outer uterine body, where 66% of cases are at visit 1, 17% at visit 2, and 17% at visit 4. These models may require additional guide curves in order to loft, or may call for an inferior-superior loft instead of left-right. The cases where the PCO is greater than posterior intrauterine diameters (UD3 and UD3b) is fairly consistent across visits. In these cases, a different measurement protocol must be used to characterize the posterior wall, as discussed in model validation in S2 Appendix.

It is observed, though not quantified, that the sagittal shape of the parametric model does not always produce a good match to the TA sagittal scan. For the anterior side, this occurs when the uterine wall is not a half ellipse. For the posterior side, this occurs when the spline does not fit the actual posterior wall shape well. The spline parameters in the models are automatically fit and no attempt is made to vary them to match individual's posterior wall shapes. This could be remedied through the use of an alternate measurement method, as discussed in S2 Appendix, or a method of capturing spline parameters from ultrasound images. Future validation of the model must be done for use in rigorous analysis of the entire gravid uterine and cervical environment. However, these low fidelity models are useful for educational and visualization purposes. Additionally, the shape and size of the lower uterine segment and cervix match well in the sagittal plane between the ultrasound and MRI-derived CAD models (S1 Fig in S2 Appendix), but improvement is still seen with an alternative measurement method. Hence initial structural analysis can be conducted of this critical *stress concentration* region [11] using a subsection of the CAD models reported here, though model accuracy is still unquantified.

## Uterine and cervical volume

The uterine volume increases over the gestational ages (Fig 8). This is observed in both the supine and standing configurations. The cervical volume does not have a clear trend of increase or decrease in volume when looking at all patients and configurations (Fig 9). Same patient, same visit uterine volume in the standing and supine positions are frequently unequal, with an average error between supine and standing of 22.1% using Eq 2.

$$\text{Error} = \frac{SupineVolume - StandingVolume}{SupineVolume} * 100, \tag{2}$$

Uterine tissue volume tends to increase at an increasing rate over the gestational ages included in this study. This result is in accordance with previous studies, which report an S-shaped curve to describe qualitatively how uterine tissue weight changes during gestation [26]. In future work, further agreement between Gillespie and parametrically estimated uterine volume can be achieved by collection of very late gestation ultrasounds to determine if tissue volume plateaus as reported. Uterine volume inconsistency between supine and standing may arise from a number of factors, including poor repeatability of uterine thickness measurements and uterine contractions occurring at time of ultrasound acquisition.

(A)

(B)

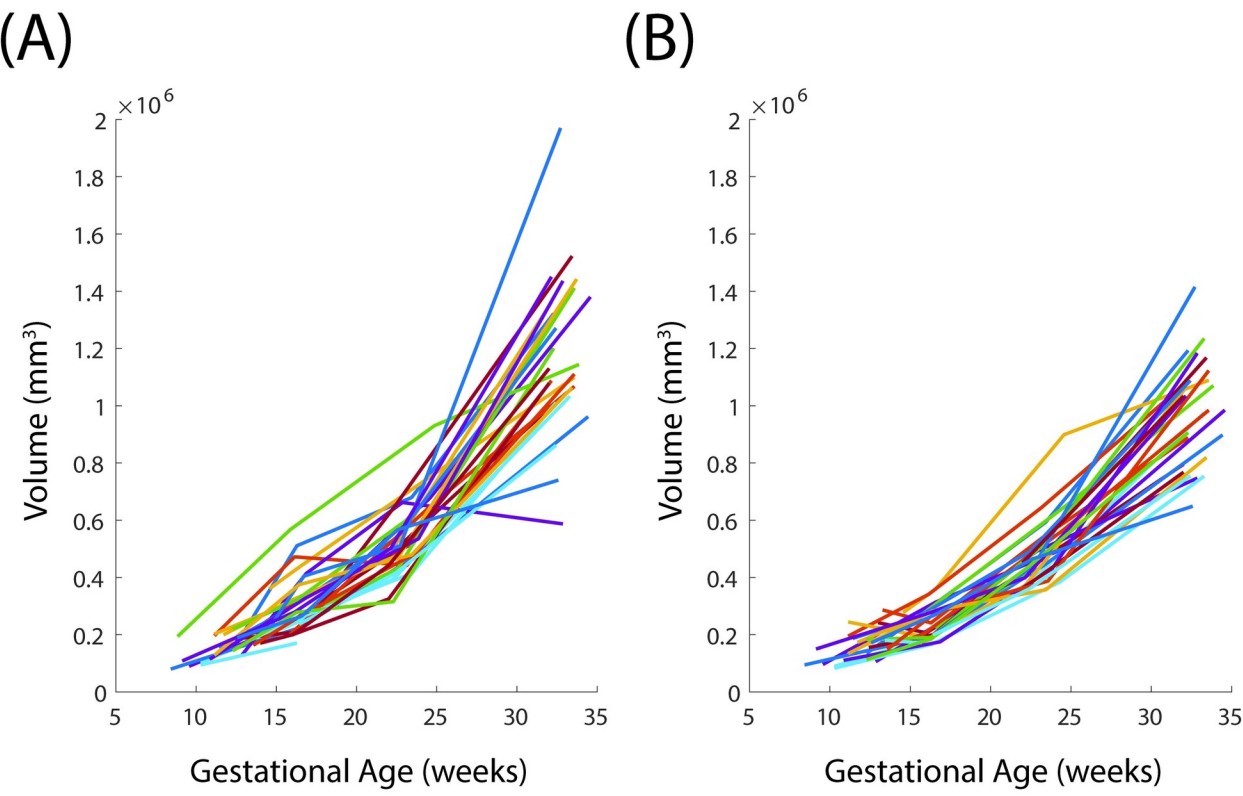

**Fig 8. Uterine volume with gestation.** Average of (A) standing and (B) supine uterine volume across three sonographers for all visits.

It is assumed, as pregnancy tissue remodeling occurs, the isthmus disappears. Due to this conclusion, it is difficult to determine if a lack cervical volume trends are a result of tissue volume changes are truly reflective of no change in tissue volumes, inconsistent cervical boundary assignment, or some combination thereof. There is currently no universally accepted method of distinguishing a boundary between the cervix and the uterus. The method used to distinguish cervical tissue from uterine tissue, described above, does not offer a rigorous delineation of tissues. While this inconsistency will not greatly influence trends in uterine volume, it has the potential to substantially skew cervical volume trends due to the smaller volume of the cervix and greater influence an equally sized error will have. It is possible that shear wave elasticity imaging could be used to better delineate between the cervical and uterine tissue, which in future work could provide a patient-specific uterocervical junction boundary.

### Comparing dimensions to previously published data

The choice of proportions to portray the uterus and cervix is informed by previous investigations of gravid geometry. Published in 1950, the last holistic study of pregnant uterine shape reports gestational-age trends of greater sagittal and transverse uterine dimensions measured from x-rays as well as uterine weights recorded retrospectively from hysterectomies executed at various stages of gravidity [26]. All data are reported either through qualitative description or graphical sketches [26]. This study concludes uterine weight increases until the 20th week of gestation, coinciding with the most rapid increase in the transverse measurement of the fundus [26]. At the 20th week of gestation the uterus is spherical and proceeds to elongate into a "cylindrical" shape until delivery [26]. A prior study of the gravid morphology in monkeys

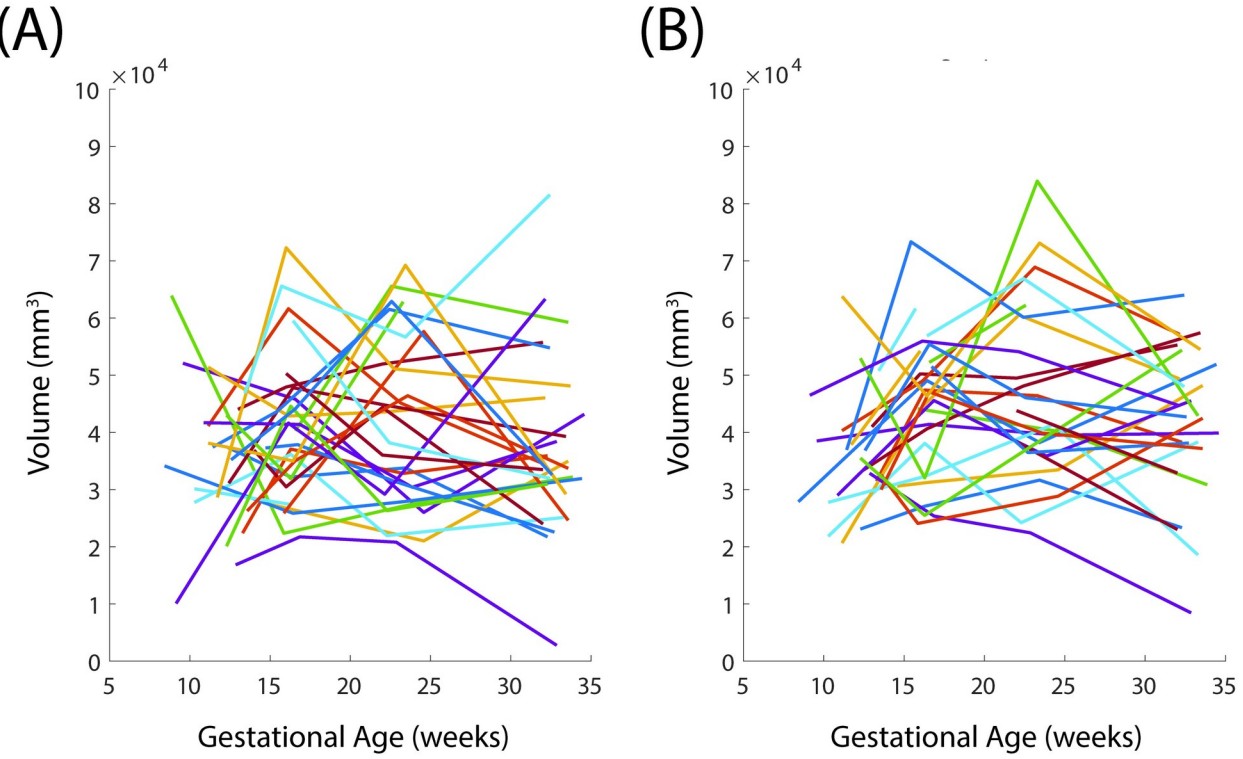

**Fig 9. Cervical volume with gestation.** Average of (A) standing and (B) supine cervical volume across three sonographers for all visits.

identified three stages of uterine development: 1. growth of the myometrium through hypertrophy accounting for the uterus's increase in weight during early pregnancy, 2. uterine growth through some hypertrophy but predominantly hyperplasia, and 3. elongation and stretching of the uterus until term [27]. In 2010, a longitudinal anatomical and cellular investigation of the myometrium in pregnant mice supported the earlier study's assertion, finding that growth in early gestation was due to hypertrophy while most growth after mid-gestation was due to hyperplasia of the smooth muscle myocytes [28].

Ultrasound investigations of the myometrium have developed differing analyses of gestational trends in thickness. Durnwald et al (2008) found a significant negative linear relationship between myometrial thickness and gravidity at the fundus, anterior wall, posterior wall, right and left-side walls, and lower uterine segment [29]. However, in an inquiry of the same five measurements, Degani et al (1998) reported only the lower uterine segment showed a significant negative correlation with gestational age [30]. Similarly, Degani et al discovered the myometrial dimensions were not significantly different from one another while Durnwald et al found the fundus was thinner than the upper uterine segment during second and third trimesters [29, 30]. Durnwald also showed multiparous women exhibited thicker uterine walls at five of the six measured sites [29]. Our own examination reviewed the myometrium at the fundus, anterior wall, side wall, and lower uterine segment.

Cervical dimensions are among the most scrutinized aspects of pregnancy, both clinically and academically. Various risk-scoring methods based on cervical diameter, dilation, length, position, and consistency have been developed from consistently found statistical correlation, though with low prognostic success [31]. Short cervical length has long been associated with PTB and the time since conception at which the measurement is taken impacts its predictive

nature [32–34]. In the first trimester, the isthmus length correlates with PTB while cervical length does not and as gestation progresses, the cervical length measurement predicts a lower risk for the patient over all [33, 35]. Recent research has also shown uterocervical angle (UCA), describing the angle at which the cervix connects with the lower uterus, also correlates with likelihood of PTB and indeed shows higher sensitivity to risk than cervical length [18]. The choice of UCA, cervical length, cervical dilation, cervical diameter and isthmus length as part of our study is based on these studies.

## Limitations

To characterize and model maternal anatomy in normal gestation, we made several simplifications to allow for implementation in the clinical setting. 2D ultrasound images allowed for data collection from more patients than if we used more detailed imaging modalities, such as 3D ultrasound or MRI. However, the 2D ultrasound images have a lower quality than other imaging techniques and preclude vision of certain anatomic features, such as the posterior uterine wall. Additionally, though the number of patients provides compelling trends in maternal anatomy evolution with gestation, a more extensive sample set would be necessary to draw population-level conclusions. The parametric modeling method, an improvement in capturing sagittal uterine shape compared to previous parametric models, is not assumed to be the most accurate method of generating patient-specific geometry. It is instead a first attempt at including more geometric sophistication. Uterine volume estimation validation is also limited, as the MRI-derived solid models used as ground-truth are from patients at greater gestational ages than those observed in this study. Thus, the novelty of the presented parametric modeling method lies in the ability to quickly generate patient-specific solid models for visualization, education, and ideation on the biomechanics of the uterus and cervix throughout gestation. It is not a rigorous basis for calculating gravid mechanical loading, though future computational studies may prove it to be so. Nevertheless, this method is foundational to our future studies of calculating stretch and stress in the pregnancy, but we have not validated its quantitative accuracy at the time of publication. While we work towards this validation, we acknowledge the importance of sharing our longitudinal measurements of the uterus and cervix in pregnancy and a straightforward method to create solid models from them.

## Conclusion

This work presents longitudinal 2D ultrasound dimension measurements which characterize the overall shape and position of the uterus and cervix, along with a framework to implement them into patient-specific parametric CAD models. In this study, the interobserver variability between measurements is explored, with measurements of intrauterine diameters, lower uterine segment thickness, anterior uterocervical angle, isthmus, and cervical length having the best repeatability. Measurements of cervical diameters, posterior cervical offset, and uterine thicknesses taken from transabdominal ultrasound show fair to poor agreement between observers. These findings are promising in refining a 2D ultrasound dimension measurement protocol that is easily integrated into clinical practice. They are also useful in establishing structural models to facilitate biomechanical calculations of tissue stress, stretch, growth and remodeling of the uterus and cervix for pregnancies at low-risk of preterm birth.

Linear mixed effect models (LMM) are calculated for all measurements, taking into account gestational age, parity, and position. Our results regarding growth of the intrauterine cavity with gestational age are intuitive, since intrauterine diameters increase with gestation to accommodate the growing fetus. The LMM models also provide insight to the effect of gravity on axial uterine shape, which becomes more oblong in the supine position compared to

standing. Parity is shown to have an effect on changes in lower uterine segment thickness and cervical length with gestation, indicating a shift in mechanical loading of the uterus and cervix in subsequent pregnancies.

The solid modeling framework is able to automatically generate patient-specific models in 91% of cases using Solidworks, a commercially available CAD software. Additional modeling frameworks will need to be developed in order to capture all uterine and cervical shapes. Uterine and cervical volume throughout gestation is estimated using the patient-specific models. Uterine volume is shown to increase with gestational age, which is in agreement with existing literature. No clear trend in cervical volume with gestational age is deduced. The current phase of the framework produces low-fidelity models appropriate for visualization and educational purposes. In future studies, the solid models will be incorporated into a finite element analysis workflow to calculate tissue stress and strain. The model's viability for finite element analysis of mechanical loading during pregnancy will be validated and necessary refinements made such that biomechanical phenomena of pregnancy can be probed. This will aid in distinguishing maternal geometry that results in a mechanically higher risk of preterm birth.

## Supporting information

**S1 Appendix.**
(PDF)

**S2 Appendix.**
(PDF)

**S3 Appendix.**
(PDF)

## Acknowledgments

The authors would like to thank Jessica Densley, Keri Johnson, and Marianne Helvey. The authors would also like to thank Jonathan Blutinger from the Creative Machines Lab at Columbia University for creating the initial parametric model of the human uterus in Solidworks.

## Author Contributions

**Conceptualization:** Veronica Over, Andrea Westervelt, Joy Vink, Timothy Hall, Helen Feltovich, Kristin Myers.

**Data curation:** Erin Marie Louwagie, Lindsey Carlson, Veronica Over, Kristin Myers.

**Formal analysis:** Lu Mao, Kristin Myers.

**Funding acquisition:** Timothy Hall, Helen Feltovich, Kristin Myers.

**Investigation:** Lindsey Carlson, Veronica Over, Kristin Myers.

**Methodology:** Erin Marie Louwagie, Veronica Over, Lu Mao, Andrea Westervelt, Kristin Myers.

**Project administration:** Helen Feltovich, Kristin Myers.

**Resources:** Helen Feltovich, Kristin Myers.

**Software:** Kristin Myers.

**Supervision:** Lindsey Carlson, Helen Feltovich, Kristin Myers.

**Validation:** Erin Marie Louwagie, Lu Mao, Kristin Myers.

**Visualization:** Erin Marie Louwagie, Shuyang Fang, Kristin Myers.

**Writing – original draft:** Lindsey Carlson, Veronica Over, Lu Mao, Kristin Myers.

**Writing – review & editing:** Erin Marie Louwagie, Helen Feltovich, Kristin Myers.

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
