## [Decision Letter · Decision Letter 0]

16 Nov 2020

PONE-D-20-33248

Longitudinal ultrasonic dimensions and parametric solid models of the gravid uterus and cervix

PLOS ONE

Dear Dr. Myers,

Thank you for submitting your manuscript to PLOS ONE. After careful consideration, we feel that it has merit but does not fully meet PLOS ONE’s publication criteria as it currently stands. Therefore, we invite you to submit a revised version of the manuscript that addresses the points raised during the review process.

An expert in the field handled your manuscript, and we are very thankful for their time and contributions. Although some interest was found in your study, some comments arose that need to be addressed in the revised manuscript.

We look forward to receiving your revised manuscript.

Kind regards,

Frank T. Spradley

Academic Editor

PLOS ONE

Reviewers' comments:

Reviewer's Responses to Questions

**Comments to the Author**

1. Is the manuscript technically sound, and do the data support the conclusions?

Reviewer #1: Yes

2. Has the statistical analysis been performed appropriately and rigorously? 

Reviewer #1: Yes

3. Have the authors made all data underlying the findings in their manuscript fully available?

Reviewer #1: Yes

4. Is the manuscript presented in an intelligible fashion and written in standard English?

Reviewer #1: Yes

5. Review Comments to the Author

Reviewer #1: Longitudinal ultrasonic dimensions and parametric solid models of the gravid uterus

and cervix

The goal of the study is to provide time-course maternal anatomy data and corresponding 3-dimensional computer aided design (CAD) models on a cohort of low-risk patients with normal singleton pregnancies. The data and models generated in this study provide the critical foundation to understand how these reproductive tissues may malfunction in pregnancy and allow for novel avenues to design biomedical devices which can be used to prevent adverse outcomes such as PTB.

Introduction

- Line 6: consider rephrasing “how well it will all go” to “any complications which may arise”

- Two time points overlap: 8-14 and 14-16 weeks. Were women in the 14th week of pregnancy allocated to both groups? As stated later in the methods section, I believe you mean the groups to be 8w0d-13w6d and 14w0d-16w6d.

Methods & Materials

- Study design is not specifically stated

- Eligibility criteria are given as women in the first trimester of pregnancy. How were the included subjects selected from the overall cohort of eligible patients?

- Inclusion criteria should be specifically stated (singleton pregnancy)

- Explain how the study size was arrived at. Was there a sample size calculation performed? If not, explain rationale for recruiting 30 subjects.

- Were fetuses affected by growth restriction or macrosomia included in the cohort? What about polyhydramnios or oligohydramnios? These changes could impact uterine size measurements.

- Why was failure to progress in labor in a prior pregnancy an exclusion criteria?

- Perhaps state that panoramic imaging is not dependent on speed of image acquisition to help orient readers to this newer imaging technique

- Is the correct terminology panoramic imaging or extended field-of-view imaging?

- Line 96: internal os (not ostia)

- Line 97: Fiji image processing package

- Line 123: mucosa

- The CL may be affected by maternal bladder filling. Was this accounted for and standardized in your protocol?

- Excellent job describing and illustrating methods used for each measurement

- Line 155: the video summary was very helpful

- Validation was performed with comparison to MRI images of uterine volume at 38 weeks. Is this an appropriate comparison given that the model was built with volumes taken at 34w or below?

Results

- Line 174: most of the measurements described in the methods for uterine dimensions/diameters were obtained using TA imaging – but here you state uterine dimensions and diameters by TV imaging. Is this a typo?

- Line 201: consider renaming this variable as endocervical canal width since you cannot confirm that you are measuring the mucous plug itself by ultrasound

- Do you propose some parameters to reduce variability in UD1 placement and thereby improve ICC for PCO?

- Do authors have any explanation for why UT1-UT3 measurements had poor ICC or how this could be optimized in future protocols?

- Line 236: It has previously been reported that…

- Line 290: It is observed that

- Line 306, 315, 320, 324: consider using weeks gestation instead of visit number

- Line 347: of 22.1%

- Line 359: a lack of cervical volume trends are truly reflective of no change in tissue volumes,…

- Line 362: Could you use elastography to better delineate between these two tissues?

Conclusion

- No changes suggested

Tables/Figures

- Figure 1 – consider adding an * to identify which structures will be evaluated by your protocol

- Table 1 – it might be important to include additional subject demographics in your table. Maternal body mass index can significantly impact ultrasound imaging quality (in particular may affect posterior uterine measurements in these women). History of prior C/S could affect the UT4 measurement.

- Figure 2 – might be helpful to clarify that the sagittal view was taken at maternal midline (perhaps include another coronal uterine outline like you do in figure 3). I believe the correct terminology is “extended field of view imaging” rather than panoramic.

- Figure 4 – again might be helpful to state sagittal view at the maternal midline

- Figure 7 – consider changing visit 1 to 8-13w, visit 2 to 14-16w etc.

6. PLOS authors have the option to publish the peer review history of their article (what does this mean?). If published, this will include your full peer review and any attached files.

Reviewer #1: No

---

## [Author Response · Author response to Decision Letter 0]

12 Dec 2020

Overview 

We thank the reviewer for the comments and the punctual response, especially during this time. We also thank the editors for giving us an opportunity to strengthen and clarify our work given the reviewers comments. We give detailed response below, and we have provided the location of manuscript updates. 

Detailed Response to Comments Introduction 

• Line 6: consider rephrasing “how well it will all go” to “any complications which may arise” 

– We have replaced ”how well it will all go” with ”any complications which may arise”. 

– Please see Line 6: any complication which may arise. 

• Two time points overlap: 8-14 and 14-16 weeks. Were women in the 14th week of pregnancy allocated to both groups? As stated later in the methods section, I believe you mean the groups to be 8w0d-13w6d and 14w0d-16w6d. 

– Thank you for catching that oversight. We replaced ”8-14, 14-16, 22-24, and 32-34 weeks” with ”8w0d-13w6d, 14w0d-16w6d, 22w0d-24w6d, and 32w0d-34w6d” 

– Please see Lines 48, 91-92, 322-324: 8w0d-13w6d, 14w0d-16w6d, 22w0d-24w6d, and 32w0d-34w6d gestation. 

Methods & Materials 

• Study design is not speciﬁcally stated 

– A study design section has now been added. 

– Please see Line 59-61: This was a longitudinal study of ultrasound dimension measurements in women at low-risk for preterm birth at 8w0d-13w6d, 14w0d-16w6d, 22w0d-24w6d, and 32w0d.34w6d gestation. 

• Eligibility criteria are given as women in the ﬁrst trimester of pregnancy. How were the included subjects selected from the overall cohort of eligible patients? 

– Flyers were given to low-risk OB clinics whose providers deliver and refer to Utah Valley Hospital in Provo. Interested patients called Lindsey Carlson and she went through the eligibility criteria with them. 

– Please see Line 63-65: Flyers describing this study were given to low-risk obstetric clinics whose providers deliver and refer to Utah Valley Hospital in Provo Utah. Interested patients called L.C.C., who reviewed eligibility criteria with them. 

• Inclusion criteria should be speciﬁcally stated (singleton pregnancy) 

– We have added ”Inclusion criteria included singleton pregnancy in the ﬁrst trimester and maternal age 18-45 years old.”. 

– Please see Line 71-72: Inclusion criteria included singleton pregnancy in the ﬁrst trimester and maternal age 18-45 years old. 

• Explain how the study size was arrived at. Was there a sample size calculation performed? If not, explain rationale for recruiting 30 subjects. 

– This research was done in tandem with research on measuring cervical sti.ness using shear wave elasticity imaging. Details on sample size rationale are published in the corresponding paper. We have added ”along with study size rationale” to indicate where study size rationale can be found. 

– Line 76-77: Details about this cohort, along with study size rationale, are published in Carlson et al. 

•Were fetuses a.ected by growth restriction or macrosomia included in the cohort? What about poly.hydramnios or oligohydramnios? These changes could impact uterine size measurements. 

–None of the pregnancies were high-risk and no pregnancy issues were found. One patient delivered preterm and was excluded from analysis. 

–Line 82-83: Of these, none of the pregnancies were considered high-risk and no pregnancy issues were found. 

•Why was failure to progress in labor in a prior pregnancy an exclusion criteria? 

–Stage 1 failure to progress in labor is basically failure of the cervix to soften and dilate. We wanted to exclude these cases because we were trying to ﬁnd pregnancies that would be as normal as possible. Because the record often doesn’t say whether failure to progress is stage 1 or 2, it was best to just assume the failure to progress was stage 1. Other reasons for cesarean section would not indicate problems in terms of cervical behavior, so those patients were not excluded. 

–Line 68-69: prior cesarean delivery for failure to progress in labour (failure of the cervix to soften/dilate) 

•Perhaps state that panoramic imaging is not dependent on speed of image acquisition to help orient readers to this newer imaging technique 

–We added ”The resulting panoramic image is not dependent on the speed of image acquisition”. 

–Please see Line 104-105: The resulting panoramic image is not dependent on the speed of image acquisition. 

•Is the correct terminology panoramic imaging or extended ﬁeld-of-view imaging? 

–The terms are similar, with panoramic being descriptive of the image itself and extended ﬁeld-of.view being descriptive of the scanning technique. We will use extended ﬁeld-of-view for consis.tency, except when discussing the style of the resulting image (1 occurrence). 

–Line 102, 106, Fig. 2 caption, Fig. 3 caption: extended ﬁeld-of-view. 

•Line 96: internal os (not ostia) 

–Replaced ”ostia” with ”os”. We have also changed all instances of inner and outer os to internal and external os for consistency. 

–Line 109: internal and external os 

•Line 97: Fiji image processing package 

–Replaced ”Fiji (ImageJ)” with ”ImageJ software” and added ”(NIH, Bethesda, MD)”. This reference style was chosen using an already published PLOS One research article as a resource (https://journals.plos.org/plosone/article?id=10.1371/journal.pone.0178488). 

–Line 110-111: ImageJ Software [10] (NIH, Besthesda, MD) 

•Line 123: mucosa 

–Replaced ”mucousa” with ”mucosa”. 

–Line 137: where the cervical mucosa ends 

•The CL may be a.ected by maternal bladder ﬁlling. Was this accounted for and standardized in your protocol? 

–All patients emptied their bladders prior to scanning. We added ”Patients emptied their bladders prior to scanning”. 

–Line 101: Patients emptied their bladders prior to scanning. 

•Excellent job describing and illustrating methods used for each measurement 

– Thank you! 

• 

Line 155: the video summary was very helpful 

– Thank you! 

• Validation was performed with comparison to MRI images of uterine volume at 38 weeks. Is this an appropriate comparison given that the model was built with volumes taken at 34w or below? 

– This is indeed a limitation to the validation of the parametric modeling method’s ability to estimate uterine volume, and we agree that a stronger case for volume estimation would be made with MRI-derived geometries from gestational ages which correspond to those used in this study. Even with limited access to MRI images stacks of pregnant anatomy, we still feel it is important to share our validation results. The following has been added to the Limitations section of the paper: ”Uterine volume estimation validation is also limited, as the MRI-derived solid models used as ground-truth are from patients at greater gestational ages than those observed in this study.” 

– Line 444-446: Uterine volume estimation validation is also limited, as the MRI-derived solid models used as ground-truth are from patients at greater gestational ages than those observed in this study. 

Results 

• Line 174: most of the measurements described in the methods for uterine dimensions/diameters were obtained using TA imaging – but here you state uterine dimensions and diameters by TV imaging. Is this a typo? 

– This is not a typo, but it is an opportunity for clariﬁcation. The overall uterine diameters, collected via TA imaging, show excellent agreement between observers, as well as dimension measurements collected via TV imaging. Dimension measurements that are not uterine diameters (uterine wall thickness, posterior cervical o.set) do NOT show excellent agreement between observers. Thus, we added ”collected via transabdominal ultrasound” after ”overall uterine diameters” to make this sentence more clear. 

– Line 188-190: Overall uterine diameters collected via transabdominal ultrasound and dimension measurements collected via transvaginal ultrasound have excellent and good agreement between observers. 

• Line 201: consider renaming this variable as endocervical canal width since you cannot conﬁrm that you are measuring the mucous plug itself by ultrasound 

– We have replaced all instances of referring to this measurement with ”inner cervical canal diam.eter” for consistency. 

– Line 135, 216, 249, 253, 649, Fig. 4 caption: inner cervical canal diameter 

• Do you propose some parameters to reduce variability in UD1 placement and thereby improve ICC for PCO? 

– We do not believe that substantial reduction of variability in the PCO measurement would be gained through additional parameters on UD1. Placing the UD1 measurement is the ﬁrst dimen.sion taken, and most measurements taken in the sagittal plane depend on its placement. That being said, UD1 has the highest ICC value of all measurements taken. In fact, all measurements that span the entire intrauterine cavity (UD1, UD23, and UD4) have the highest ICC values of the 16 parameters measured. All measurements that rely on UD1, besides the PCO, have excel.lent or good agreement. The PCO is unique in that it relies on the internal cervical os, which is di.cult to see in TA sagittal scans. We believe this is the primary cause of the PCO having poor agreement. We have changed the verbiage to reﬂect this belief. 

– Line 241-247: The poor agreement between observers for posterior cervical o.set (PCO) measure.ments may also be attributed to variable inferior-superior intrauterine diameter (UD1) placement, as it is used as the end point for this dimension. Additionally, identiﬁcation of the internal cervi.cal os is often di.cult in TA sagittal scans (this is why the clinical gold standard for measurement of the cervix is TV), most likely the primary cause for poor agreement on the PCO parameter. 

• Do authors have any explanation for why UT1-UT3 measurements had poor ICC or how this could be optimized in future protocols? 

– We have added ”The ICC value for uterine thickness measurements captured using TA transducers could be improved by taking taking individual, zoomed in ultrasound images of the uterine wall at the fundus, anterior wall, and left or right wall.” 

– Line 236-239: The ICC value for uterine thickness measurements captured using TA transducers could be improved by taking taking individual, zoomed in ultrasound images of the uterine wall at the fundus, anterior wall, and left or right wall. 

•Line 236: It has previously been reported that. . . 

–We have replaced ”It has previously been reported” with ”Previous studies report” 

–Line 255-256: Previously studies report isthmus length (IS) and anterior uterocervical angle (AUCA) to be considered repeatable measurements 

•Line 290: It is observed that 

–We have removed ”it is observed” and instead written ”uterine shape does change” 

–Line 309-310: However, uterine shape does change with position 

•Line 306, 315, 320, 324: consider using weeks gestation instead of visit number 

–We have taken the opportunity to remind the reader of the visit gestational windows in line 306, but think that repetition throughout the rest of the section is unnecessary. 

–Line 324-326: For visits 1-3 (8w0d-13w6d, 14w0d-16w6d, and 22w0d-24w6d gestation), the spline method better represents the posterior uterine wall, and for visit 4 (32w0d-34w6d) the quarter ellipse method is a better method 

• Line 347: of 22.1% 

–We have replaced ”or” with ”of” 

–Line 366: average error between supine and standing of 22.1% 

•Line 359: a lack of cervical volume trends are truly reﬂective of no change in tissue volumes,. . . 

–We have added ”are truly reﬂective of no change in tissue volumes” 

–Line 378-380: it is di.cult to determine if a lack cervical volume trends are a result of tissue volume changes are truly reﬂective of no change in tissue volumes, inconsistent cervical boundary assignment, or some combination thereof 

•Line 362: Could you use elastography to better delineate between these two tissues? 

–Yes, absolutely, and it is currently something we are investigating in terms of trying to identify the isthmus using elasticity imaging. 

–Line 386-388: It is possible that shear wave elasticity imaging could be used to better delineate between the cervical and uterine tissue, which in future work could provide a patient-speciﬁc ute.rocervical junction boundary. 

Tables/Figures 

•Figure 1 – consider adding an * to identify which structures will be evaluated by your protocol 

–An * was added to uterine fundus, uterus, lower uterine segment, cervix, internal os, cervical canal, and external os. In the ﬁgure description, ”Asterisks (*) indicate structures evaluated in the protocol” was added. We have also changed all instances referring to the internal os, external os, and exocervix to use this verbiage, rather than inner os, outer os, and ectocervix for consistency. 

–Fig. 1 caption: Asterisks (*) indicate structures evaluated in the protocol 

–Line 244, 541, Fig. 2 caption, Fig. 5 caption: internal, external, OR exocervix 

•Table 1 – it might be important to include additional subject demographics in your table. Maternal body mass index can signiﬁcantly impact ultrasound imaging quality (in particular may a.ect posterior uterine measurements in these women). History of prior C/S could a.ect the UT4 measurement. 

–Three patients had a previous c-section, which has been included in Table 1. There were no BMIs larger than 30. 

–Table 1 caption: There were no patients with a BMI larger than 30. 

•Figure 2 – might be helpful to clarify that the sagittal view was taken at maternal midline (perhaps include another coronal uterine outline like you do in ﬁgure 3). I believe the correct terminology is “extended ﬁeld of view imaging” rather than panoramic. 

–A coronal view of the uterus with a dashed line from inferior to superior along the maternal midline was added. In the ﬁgure description, ”coronal uterine outline with the ultrasound sweep location shown with a dashed line” was added. 

–Fig. 2 caption: coronal uterine outline with the ultrasound sweep location shown with a dashed line, and a representative illustration of measurements taken from ultrasounds of the uterus and cervix in the sagittal view 

•Figure 4 – again might be helpful to state sagittal view at the maternal midline 

–In the ﬁgure description, ”at the maternal midline” was added. 

–Fig. 4 caption: representative illustration of measurements taken from ultrasounds of the cervix and lower uterine segment in the sagittal view at the maternal midline 

•Figure 7 – consider changing visit 1 to 8-13w, visit 2 to 14-16w etc. 

–In the ﬁgure caption, we added ”for visits 1-4 (8w0d-13w6d, 14w0d-16w6d, 22w0d-24w6d, and 32w0d-34w6d gestation)”. 

–Fig. 7 caption: for visits 1-4 (8w0d-13w6d, 14w0d-16w6d, 22w0d-24w6d, and 32w0d-34w6d ges.tation)

---

## [Decision Letter · Decision Letter 1]

26 Dec 2020

Longitudinal ultrasonic dimensions and parametric solid models of the gravid uterus and cervix

PONE-D-20-33248R1

Dear Dr. Myers,

We’re pleased to inform you that your manuscript has been judged scientifically suitable for publication and will be formally accepted for publication once it meets all outstanding technical requirements.

Kind regards,

Frank T. Spradley

Academic Editor

PLOS ONE

Reviewers' comments:

Reviewer's Responses to Questions

**Comments to the Author**

1. If the authors have adequately addressed your comments raised in a previous round of review and you feel that this manuscript is now acceptable for publication, you may indicate that here to bypass the “Comments to the Author” section, enter your conflict of interest statement in the “Confidential to Editor” section, and submit your "Accept" recommendation.

Reviewer #1: All comments have been addressed

2. Is the manuscript technically sound, and do the data support the conclusions?

Reviewer #1: Yes

3. Has the statistical analysis been performed appropriately and rigorously? 

Reviewer #1: Yes

4. Have the authors made all data underlying the findings in their manuscript fully available?

Reviewer #1: Yes

5. Is the manuscript presented in an intelligible fashion and written in standard English?

Reviewer #1: Yes

6. Review Comments to the Author

Reviewer #1: (No Response)

7. PLOS authors have the option to publish the peer review history of their article (what does this mean?). If published, this will include your full peer review and any attached files.

Reviewer #1: No

---

## [Editor Report · Acceptance letter]

19 Jan 2021

PONE-D-20-33248R1 

Longitudinal ultrasonic dimensions and parametric solid models of the gravid uterus and cervix  

Dear Dr. Myers:

I'm pleased to inform you that your manuscript has been deemed suitable for publication in PLOS ONE. Congratulations! Your manuscript is now with our production department. 

Kind regards, 

on behalf of

Dr. Frank T. Spradley 

Academic Editor

PLOS ONE